# An Overview of Ovarian Calyx Fluid Proteins of *Toxoneuron nigriceps* (Viereck) (Hymenoptera: Braconidae): An Integrated Transcriptomic and Proteomic Approach

**DOI:** 10.3390/biom13101547

**Published:** 2023-10-19

**Authors:** Rosanna Salvia, Carmen Scieuzo, Andrea Boschi, Marco Pezzi, Michele Mistri, Cristina Munari, Milvia Chicca, Heiko Vogel, Flora Cozzolino, Vittoria Monaco, Maria Monti, Patrizia Falabella

**Affiliations:** 1Department of Sciences, University of Basilicata, Via dell’Ateneo Lucano 10, 85100 Potenza, Italy; r.salvia@unibas.it (R.S.); andrea.boschi@unibas.it (A.B.); 2Spinoff XFlies s.r.l., University of Basilicata, Via dell’Ateneo Lucano 10, 85100 Potenza, Italy; 3Department of Chemical, Pharmaceutical and Agricultural Sciences, University of Ferrara, Via Luigi Borsari 46, 44121 Ferrara, Italy; marco.pezzi@unife.it (M.P.); michele.mistri@unife.it (M.M.); cristina.munari@unife.it (C.M.); 4Department of Life Sciences and Biotechnology, University of Ferrara, Via Luigi Borsari 46, 44121 Ferrara, Italy; milvia.chicca@unife.it; 5Department of Entomology, Max Planck Institute for Chemical Ecology, Hans-Knoll-Straße 8, D-07745 Jena, Germany; hvogel@ice.mpg.de; 6Department of Chemical Sciences, University of Naples Federico II, 80126 Naples, Italy; flora.cozzolino@unina.it (F.C.); monacovi@ceinge.unina.it (V.M.);; 7CEINGE Advanced Biotechnologies Franco Salvatore, 80145 Naples, Italy

**Keywords:** *Heliothis virescens*, ovarian calyx fluid proteins, proteomic and transcriptomic approach, *Toxoneuron nigriceps*

## Abstract

The larval stages of the tobacco budworm, *Heliothis virescens* (Fabricius) (Lepidoptera: Noctuidae), are parasitized by the endophagous parasitoid wasp, *Toxoneuron nigriceps* (Viereck) (Hymenoptera: Braconidae). During the injections of eggs, this parasitoid wasp also injects into the host body the secretion of the venom gland and the calyx fluid, which contains a polydnavirus (*T. nigriceps* BracoVirus: *Tn*BV) and the Ovarian calyx fluid Proteins (OPs). The effects of the OPs on the host immune system have recently been described. In particular, it has been demonstrated that the OPs cause hemocytes to undergo a number of changes, such as cellular oxidative stress, actin cytoskeleton modifications, vacuolization, and the inhibition of hemocyte encapsulation capacity, which results in both a loss of hemocyte functionality and cell death. In this study, by using a combined transcriptomic and proteomic analysis, the main components of *T. nigriceps* ovarian calyx fluid proteins were identified and their possible role in the parasitic syndrome was discussed. This study provides useful information to support the analysis of the function of ovarian calyx fluid proteins, to better understand *T. nigriceps* parasitization success and for a more thorough understanding of the components of ovarian calyx fluid proteins and their potential function in combination with other parasitoid factors.

## 1. Introduction

Insects represent the widest group of living organisms, and they occupy almost every ecological niche [1]. Their evolutionary success is the result of a long adaptive radiation that has led to the development of mechanisms for the exploitation of the most diverse substrates [2].

Among these mechanisms, those involved in antagonistic insect/insect relations are the most complex and diversified, from the relatively simple trophic relationship between predatory insects and their hosts, to the more complex biological and physiological relationships between parasitoids and their hosts [3,4]. Parasitoids exploit the host during the juvenile stages and then they bring it to death, so they are able to coexist and interact with the host for longer or shorter periods [5]. Parasitoids are able to regulate host physiology, development, and reproduction, with the sole aim of building the best environmental and nutritional conditions for the survival and the development of their progeny [6]. The study of the physiological and molecular mechanisms at the base of these antagonistic relationships has shown that the host–parasitoid interactions represent an interesting opportunity for isolating genes and molecules with potential insecticidal activity, which can be used for the development of innovative technologies in the biocontrol of harmful insects [7,8].

From an evolutionary point of view, the mechanisms adopted by wasps to ensure the successful development of their progeny are really sophisticated [9]. Endo- and ectoparasitoids have both evolved several strategies to manipulate host physiology in order to create a favorable environment for the development of their progeny, damaging that of the host insect [10,11,12,13,14]. The success of parasitism usually depends on factors that adult wasps inject during oviposition. These factors could be of maternal origin, such as venom, ovarian calyx fluid, polydnavirus, or virus-like particles, or of embryonic origin, such as teratocytes, which are cells deriving from embryonic serosa. They often induce complex physiological alterations in hosts that ensure parasitoid development. The alterations that parasitoids induce are currently referred to as host regulation factors [15]. Many studies on parasitoid physiology focused on the identification of these factors and their effects on the host [11,16,17,18]. Among these factors, secretions (venom and ovarian calyx fluid) of female endoparasitoids, injected during oviposition, play a key role in the induction of major alterations observed in parasitized hosts. The venom, produced by glands associated with the female reproductive system, is a complex mixture composed mostly of proteinaceous compounds [19]. Venom from ectoparasitoid wasps has mainly been associated with host paralysis [20] and developmental block [20,21]. In endoparasitoids, venom is often responsible for the suppression of the host immune response and developmental changes, together with calyx fluid components [22,23]. The calyx fluid contains the ovarian proteins (OPs) and in some cases a polydnavirus (PDV) or virus-like particles (VLP) containing protein virulence factors [24,25]. The OPs, together with venom, are responsible for quick immunosuppression right after oviposition. Polydnaviruses (PDVs) are obligate symbionts integrated as a provirus in the genome of endoparasitoids, where they replicate in the ovary to produce free viral particles. After oviposition, PDVs infect host tissues and express gene products responsible for several alterations of the host physiology [11,22]. The success of parasitism is also assured by embryonic factors, such as the teratocytes [26]. In the host–parasitoid system, *Heliothis virescens*–*Toxoneuron nigriceps*, venom, teratocytes, and some PDV genes have already been studied [27,28,29,30,31,32,33,34] and recently the effects of the OPs on the host immune system have been reported [35,36]. The OPs have been shown to induce several alterations on hemocytes, including cellular oxidative stress, modifications of actin cytoskeleton, vacuolization, and the inhibition of encapsulation capacity of hemocytes, thus inducing both a loss of hemocyte functionality and cell death [36]. Overall, OPs, in association with PDV and venom, contribute to the evasion of the host immune response by *Toxoneuron nigriceps* (Viereck) (Hymenoptera: Braconidae). Moreover, OPs’ HPLC fractions have been tested and, for two fractions, the same effects have been observed as shown in the previous work [35]. In addition, we identified the proteins of these two fractions through a combination of transcriptomic and proteomic approaches, resulting in the identification of eight proteins that could be involved in the observed alterations of the host hemocytes.

Here, using a combined transcriptomic and proteomic approach, we identified the major components of *T. nigriceps* ovarian calyx fluid proteins. Transcriptomic information deriving from the *de novo* transcriptome analysis provided an overall picture of the putative proteins and their molecular functions, biological processes, and putative cellular compartments. Proteomic analysis was carried out on the components of the ovarian fluid, fractionated by SDS-PAGE electrophoresis, and analyzed using mass spectrometry (nanoLC-MS/MS). The comparison between translated transcriptomic and proteomic data allowed us to identify the expressed OPs.

This study provides useful information for the investigation of the role of ovarian calyx fluid proteins in *T. nigriceps*’ parasitization success.

## 2. Materials and Methods

### 2.1. Insect Rearing

*Toxoneuron nigriceps* and *Heliothis virescens* (Fabricius) (Lepidoptera: Noctuidae) were reared according to a previous protocol [37]. Briefly, larvae of *H. virescens* were reared on an artificial diet based on bean flour and wheat germ [38]. Unparasitized and parasitized *H. virescens* larvae were maintained in plastic vials (60 × 20 mm) with diet, with one larva per vial to prevent cannibalism, at 29 ± 1 °C, while *T. nigriceps* adults were fed daily with a 40% honey/water solution and maintained in cages at 25 ± 1 °C. Both species were kept at a relative humidity of 70 ± 5% and a 16 h:8 h (light: darkness) photoperiod. Host larvae were parasitized at the third day of the last (fifth) instar by placing larvae with a single mated female wasp in a Petri dish (55 × 15 mm). Parasitoid females were allowed to lay only one egg per host larva. Then, parasitized hosts were placed individually in vials until the parasitoid egression from the host.

### 2.2. Calyx Fluid Collection, Protein Purification, and RNA Extraction

To collect the calyx fluid containing the Ovarian Proteins (OPs) and the polydnavirus of *T. nigriceps* (*Tn*BV), two-week-old adult females of *T. nigriceps* were employed. From each female, anesthetized on ice for 10 to 15 min and immersed in a physiological solution of 1× Phosphate-Buffered Saline (PBS), the entire reproductive apparatus was removed, as described by Salvia et al. [36]. To avoid any hemolymph contamination, the reproductive apparatus was washed several times with 1× PBS. The isolated ovaries were placed in a drop of 20 µL of 1× PBS at 4 °C and the ovarian calyx were dissected to allow the flow of the calyx fluid. The ovarian calyx tissue was placed in a 1.5 mL centrifuge tube (Eppendorf, Hamburg, DE, USA) containing TRI Reagent (Sigma, St. Louis, MO, USA) and stored at −80 °C for RNA extraction and subsequent transcriptomic analysis [23], while the fluid was transferred into a 1.5 mL tube (Eppendorf) and centrifuged at 2000× *g* for 5 min at 4 °C to remove debris. As previously described [35,36], the supernatant, containing ovarian calyx fluid proteins and the polydnavirus, was further purified with 0.45 μm Millex PVDF filters followed by washing 3 times with 1 mL of 1× PBS. To separate the polydnavirus, a centrifugation was carried out at 30,000× *g* for 1 h at 4 °C and subsequently the supernatant containing the ovarian calyx fluid proteins was concentrated by ultrafiltration in a column with cut-off 3000 MW, by centrifuging the sample at 3000× *g* until a volume of around 500 μL was obtained.

As previously described [23,35], the TRI Reagent (Sigma, St. Louis, MO, USA) was used to extract total RNA from around 80–100 ovarian calyxes, according to the manufacturer’s instructions (Sigma, St. Louis, MO, USA). To remove any DNA contamination, a DNase (Turbo DNase, Ambion Austin, TX, USA) treatment was carried out. After removing the DNase enzyme, the RNA was purified using the Rneasy MinElute Clean-up Kit (Qiagen, Venlo, The Netherlands) according to the manufacturer’s instructions and eluted in 20 µL of RNA Storage Solution (Ambion, Austin, TX, USA). Nanodrop ND1000 spectrophotometer was used to measure the RNA amount and Agilent 2100 Bioanalyzer (Agilent Technologies, Palo Alto, CA, USA) was used to verify the RNA integrity.

### 2.3. RNASeq Data Generation and De Novo Transcriptome Assembly

Tissue-specific transcriptomic (RNAseq) data from different *T. nigriceps* tissues (venom glands, teratocytes, and ovarian calyx) were generated as previously described [23]. Briefly, tissue-specific transcriptome sequencing of the RNA samples was performed with poly(A)þ enriched mRNA fragmented to an average of 150 nucleotides. Sequencing was carried out by the Max Planck Genome Center (http://mpgc.mpipz.mpg.de/home/, accessed on 24 October 2015) using standard TruSeq procedures on an Illumina HiSeq2500 sequencer, generating approximately 40 Mio paired-end (2 × 100 bp) reads for each of the tissue samples. Quality control measures, including the filtering of high-quality reads based on the score given in fastq files, the removal of reads containing primer/adaptor sequences, and the trimming of the read length, were carried out using CLC Genomics Workbench v7.1 (http://www.clcbio.com, accessed on 24 October 2015). The *de novo* transcriptomes assembly was carried out with the same software using RNAseq data (30–90 Mio paired-end reads). The sequence reads deriving from the different tissues were combined in a joint transcriptome assembly and separately assembled to obtain tissue-specific transcriptomes, and the presumed optimal consensus transcriptome was selected as described in Vogel et al. [39]. All obtained sequences (contigs) were used as the query for a BLASTx search [40] in the ‘National Center for Biotechnology Information’ (NCBI) non-redundant (nr) database, considering all hits with an e-value < 10^−5^. The transcriptome was annotated using BLAST, Gene Ontology, and InterProScan searches using BLAST2GO PRO v2.6.1 (www.blast2go.de, accessed on 10 November 2015) [41]. A custom-made protein database was created using the previously assembled and annotated *T. nigriceps* ovarian calyx transcriptome [23]. The six reading frames of the 24,760 contigs derived from the transcriptome were translated in their respective amino acid sequences using SEQtools software (2012 version), obtaining 148,560 sequences. The “Ovarian calyx fluid proteins *T. nigriceps* database” provides useful information for protein identification, combining transcriptomic and proteomic data.

### 2.4. Protein Separation and Identification

Protein of the ovarian calyx fluid extract of *T. nigriceps* was fractionated by SDS-PAGE, and a sample deriving from 30 females was loaded. The gel was stained with GelCode™ Blue Safe Protein Stain (Thermo Fisher Scientific, Waltham, MA, USA) and destained with Milli-Q water. A total of 13 bands were cut and in situ hydrolyzed by trypsin [42,43]. Peptide mixtures were extracted in 0.2% HCOOH and ACN and vacuum dried by a SpeedVac System (Thermo Fisher Scientific).

Each peptide mixture was dissolved in 10 μL of 0.2% HCOOH (Sigma Aldrich) and analyzed using nanoLC-MS/MS on an LTQ Orbitrap mass spectrometer coupled to the nanoACQUITY UPLC system (Waters, Sesto San Giovanni, Milan, Italy). Samples were first concentrated on a C18 capillary reverse-phase pre-column (20 mm, 180 μm, 5 μm) and then fractionated on a C18 capillary reverse-phase analytical column (250 mm, 75 μm, 1.8 μm) working at a flow rate of 300 nL/min, using a linear two-step gradient.

The first step was from 5% B to 35% B in 70 min, and the second from 35% B to 80% B in 5 min, with the following eluent B (0.2% formic acid in 95% acetonitrile) and A (0.2% formic acid and 2% acetonitrile in LC-MS/MS grade water, Merck Millipore, Burlington, MA, USA). The MS/MS analyses were performed using Data-Dependent Acquisition (DDA) mode, after one full MS scan (mass range from 400 to 1800 *m*/*z*). The 5 most abundant ions were selected for the MS/MS scan events, applying a dynamic exclusion window of 40 s. The peak list generated was uploaded in Mascot software v2.4.0 and research was performed against the in silico peak list obtained from the custom-made database “Ovarian calyx fluid proteins *T. nigriceps* database” that we previously generated [23,35], since for our model system no sequences were available in the database. The parameters for protein identification were “trypsin” as the enzyme (one missed cleavage), “carbamidomethyl” as a fixed modification, “oxidation of Met” and “pyro-Glu at N-term if Gln” as variable modifications, 0.6 Da as MS/MS tolerance, and 10 ppm as peptide tolerance. The scores threshold of matches for MS/MS data was fixed at least 8 for all peptides, as suggested by the Mascot algorithm. Functional protein characterization was performed by aligning the frame of identified sequences versus all proteins of the eukaryotic database by the BLAST tool of Uniprot open source. The presence of the signal peptide was verified with the software Signal P 6.0 (http://www.cbs.dtu.dk/services/SignalP/, accessed on 9 July 2023).

## 3. Results

The *de novo* transcriptome of *T. nigriceps* ovarian calyx was built containing 24,760 contigs (Appendix A) with a minimum contig size of 250 bp and a maximum contig length of 27,912 bp, as previously described [35].

To identify similarities with annotated proteins, the contig sequences of the *de novo* transcriptome of *T. nigriceps* ovarian calyx were searched using the BLASTx algorithm against a non-redundant (nr) NCBI protein database (Appendix A). The species distribution of the top BLAST hit against the nr database showed that the majority of obtained top hits matched against *Microplitis demolitor* Wilkinson (Hymenoptera: Braconidae) (Figure 1).

For functional annotation, all sequences were subjected to gene ontology (GO) analysis in Blast2GO, classifying contigs into three GO categories: cellular components, molecular functions, and biological processes. The most prevalent category of the GO cellular component was represented by membranes (27.53%) (Figure 2a—GO level 3) and integral components of membranes (38.04%) (Figure 2d—GO level 4). The most prevalent molecular function involved proteins binding organic cyclic compounds (18.30%), followed by heterocyclic binding proteins (18.25%) (Figure 2b—GO level 3). The most abundant molecular function involved proteins binding nucleic acid (18.72%) (Figure 2e—GO level 4). The most abundant category of GO biological processes was composed of proteins involved in organic substance metabolic processes (14.39%) (Figure 2c—GO level 3), especially metabolic processes of macromolecules (10.73%) (Figure 2f—GO level 4).

The enzyme code distribution shows that the most abundant families of enzymes were hydrolases and transferases (Figure 3).

Among the transcriptome contigs, we also identified *T. nigriceps* polydnavirus-related transcripts (Appendix A). In total, we found 88 contigs, of which 16 contigs belonged to the wasp and were annotated as viral capsid proteins. Moreover, surprisingly, we also found 72 sequences of viral genes normally expressed into the host. In particular, 30 contigs included sequences annotated as viral genes encoding for some of the known *Tn*Bv transcripts, whose sequences were deposited in the database (PTP1, PTP2, PTP3, PTP4, PTP5, PTP6, PTP12, *Tn*BV1, *Tn*BV2, *Tn*BVank1, *Tn*BVank2, and *Tn*BVank3). The remaining 42 contigs were annotated as sequences similar to other polydnavirus genes different from *Tn*BV.

Using a combined transcriptomic and proteomic approach, a total of 105 ovarian calyx fluid proteins of *T. nigriceps* were identified. The ovarian calyx fluid proteins were fractionated by SDS-PAGE, and 13 lines were excised from the gel (Figure 4) and in situ hydrolyzed.

The peptide mixtures were analyzed through LC-MS/MS analysis with an LTQ-Orbitrap XL instrumentation. The mgf data obtained were processed with Mascot software by using, as a database, the “Ovarian calyx fluid proteins *T. nigriceps* database” [35] consisting of putative protein sequences deduced from the transcriptomic analysis of *T. nigriceps* and present in the form of contig. Associated with each contig, six putative protein sequences are reported, each corresponding to a single reading frame.

The amino acid sequences identified by Mascot allowed the identification of the ovarian calyx fluid proteins and their reading frames, and then the specific sequences were used for alignment with homologous proteins belonging to organisms phylogenetically close to the parasitoid, by using two different pieces of software: the BLASTx (https://blast.ncbi.nlm.nih.gov/Blast.cgi?PROGRAM=blastx&PAGE_TYPE=BlastSearch&LINK_LOC=blasthome, accessed on 5 October 2023) and the BLASTp (https://blast.ncbi.nlm.nih.gov/Blast.cgi?PROGRAM=blastp&PAGE_TYPE=BlastSearch&LINK_LOC=blasthome, accessed on 5 October 2023).

Appendix A reports the 105 identified proteins in the 13 lines, including information regarding the contig and the frame identified, the number of the total peptides, and the sequence coverage of the contig and the name of the associated protein obtained after the alignment. Appendix A reports all the other information, including the sequence of identified peptides, the Mascot score, the *m*/*z* observed, the frame number of the transcriptomic sequence obtained with SEQtools matching with the LC-MS/MS, the start and the end of each identified peptide, the number of peptides identified for each protein, the contig code, the amino acid sequence frame (the tryptic peptides experimentally observed by LC-MS/MS analysis, used for the sequence coverage calculation, are reported in red), the sequence molecular weight, the percentage of mass peptide sequence coverage, the protein name, the query cover and the identity percentage %, and the *E*-value of the candidate from BLASTp and from BLASTx alignment.

## 4. Discussion

The transcriptomic analysis of ovarian calyx tissue allowed the identification of 24,760 transcripts. Most of them encode structural proteins (such as membrane proteins or actin), and proteins involved in cellular metabolism, DNA replication, transcription, and protein synthesis. Moreover, *T. nigriceps* polydnavirus-related transcripts were identified (Appendix A). Out of the 88 contigs, 16 belong to the wasp and are annotated as viral capsid proteins, but surprisingly we found also 72 sequences of viral genes normally expressed in the host. Thirty contigs included sequences annotated as viral genes encoding for some of the known *Tn*Bv transcripts, whose sequences are deposited in the database, as viral ankyrins (*Tn*BVank1, *Tn*BVank2 and *Tn*BVank3), protein tyrosine phosphatase (PTP1, PTP2, PTP3, PTP4, PTP5, PTP6, PTP12), *Tn*BV1, and *Tn*BV2. We cannot exclude that the part of the sequences in each contig not related to *Tn*Bv genes did not belong to regions of the viral DNA, since data of the full genome sequencing of *T. nigriceps* polydnavirus have not yet been published. The remaining 42 contigs were annotated as sequences similar to other polydnavirus genes different from *Tn*BV. It is plausible to believe that they are contigs belonging to *Tn*BV not yet available in the databases.

However, the presence of contigs related to the polydnavirus genome could be due to residual polydnavirus DNA present in the RNAseq libraries, even if RNA was treated with DNase; indeed, the RNAseq technique is so deep that it can detect very tiny amounts of nucleic acids. To verify this hypothesis, we compared the contigs related to the polydnavirus genome with the sequences of the deposited *Tn*BV genes. Indeed, while genes encoding for PTPs and ankyrins do not contain introns [30,31], for the *Tn*BV1 gene, the structure of the gene available in Varricchio et al. [34] reports the presence of an intron, which we found in the contig, confirming that these are fragments of the viral genome that could derive from contaminating viral particles or from the wasp genome during the mechanism of replication and assembly into nucleocapsids that occurs in calyx cells [18].

Combining transcriptomic data of ovarian calyx and proteomic data of ovarian calyx fluid we identified 105 proteins belonging to the ovarian calyx fluid. Among the identified proteins, nine were related to the polydnavirus, since they had high similarity with other PDV, corresponding to the following contigs: T_C2669, T_C1672, T_C1283, T_C3680, T_C2407, T_C1206, T_C1259, T_C216, T_C387. Moreover, an egg yolk protein, the major royal jelly protein 5 (T_C1629), and an ovalbumin protein (T_C3785) (Appendix A) were identified. To verify the presence of parameters indicative of the secretion mechanism, we searched for the presence of the signal peptide.

Among the remaining 94 proteins, 28 had signal peptides and cleavage sites. Among the 66 contigs lacking the signal peptide, 20 had a complete sequence and 46 had an incomplete sequence at 5′ (Appendix A). The 46 incomplete proteins were aligned, using BLAST hits from the NCBI non-redundant (nr) database, with homologous proteins with a high percentage of similarity and query cover, and we found that 12 proteins were similar to proteins with signal peptide, leading us to reasonably think that these proteins are secreted. Regarding the remaining 20 proteins with a complete sequence and without signal peptide, and the 34 with an incomplete sequence at 5′, whose most similar proteins lacked the signal peptide, we cannot exclude that they are proteins secreted in the calyx fluid, as it is known that some proteins use unconventional secretion mechanisms such as, for example, the vesicular traffic [26]. This has been found for some proteins of parasitic origin; an example is the enolase, known as multifunctional protein, and it has been reported that, in various systems, it is released outside the cells by vesicular trafficking [26].

Among the identified proteins, eight had already been described in a previous work [36], in which the effects of HPLC fractions deriving from OPs were studied *in vitro* on *H. virescens* hemocytes after 2h of treatment: oxidative stress, vacuolization, cytoskeletal damages, and the inhibition of encapsulation were observed. Out of the 28 HPLC fractions, two (#22 and #26) apparently showed a reduction in cell viability and then the subsequent analyses focused on these two fractions, studying their role in hemocyte morphology and functional changes, and also trying to identify the proteins contained in these fractions. In HPLC fraction #22, mitotic spindle organizing 2-like protein and FK506-binding protein 59 were identified. In HPLC fraction #26, glyceraldehyde-3-phosphate dehydrogenase, phosphoglycerate mutase, glutathione transferase, proliferating cell nuclear antigen, apolipophorin-III, and Cu/Zn−superoxide dismutase were identified.

Many of the identified proteins in the present study could play a crucial role in the parasitic syndrome, although not immediately after oviposition, unlike the proteins identified in the two HPLC fractions, which have been shown to play an essential role in the early suppression of the host immune response.

The major components of *T. nigriceps* ovarian calyx fluid proteins were hydrolases and transferases. The hydrolase class has already been reported as a common component in the venoms of several species of parasitoids [44]. Among the hydrolases in the *T. nigriceps* OPs, esterase, phosphatase, aminopeptidase (cytosol aminopeptidase, dipeptidyl peptidase, prolyl endopeptidase), protease and nuclease (ribonuclease Oy, DNA endonuclease), glycosidases, metalloprotease, and enolase were identified. All these proteins are also involved in the metabolic functions of the cells. Metalloproteases have been found in the venom of several parasitoids, including *T. nigriceps* venom [23]. Metalloproteases could have a triple role, and indeed could be involved in host immune suppression, in the control of host development, and in the degradation of the extracellular matrix to provide nutritional elements for the parasitoid offspring [23]. It has been reported that a metalloprotease of the enterobacterium *Photorhabdus luminescens* is responsible for the cutting of immune proteins in the host *Manduca sexta* Linnaeus (Lepidoptera: Sphingidae) hemolymph, blocking the host immune response [45]. Moreover, it has been shown that a metalloprotease deriving from the parasitoid wasp *Eulophus pennicornis* Nees (Hymenoptera: Eulophidae) induced a delay in development with a reduction in growth into the host larvae of *Lacanobia oleracea* Linnaeus (Lepidoptera: Noctuidae) [46]. Considering the toxic effects of metalloprotease, the insecticidal activity of a metalloprotease produced by *Photorhabdus luminescens* strain 0805-P5G was successfully tested against *Galleria mellonella* (Linnaeus) (Lepidoptera: Pyralidae) and *Plutella xylostella* (Linnaeus) (Lepidoptera: Plutellidae) [47].

An important role in parasitic syndrome is played by serpin proteins involved in numerous biological processes, including the regulation of polyphenol oxidase activation. Serine proteases were also found in *T. nigriceps* venom, in the ovarian calyx fluid proteins of the endoparasitoid *Cotesia chilonis* (Matsumura) (Hymenoptera: Braconidae) [25], and in other parasitoid venoms [23,48,49]. These proteins can play a crucial role in host regulation, inhibiting melanization in the host hemolymph by blocking the phenoloxidase cascade [50]. It has been demonstrated in *Cotesia rubecula* (Marshall) (Hymenoptera: Braconidae) that a serine protease inhibits melanization, presumably by competing with host serine protease homologs for binding the prophenoloxidase that remains uncleaved in the hemolymph [50,51]. Choo and colleagues [52] showed that a serine protease found in the bee venom seemed to be involved in the hyperactivation of prophenoloxidases, resulting in a lethal melanization response in the target insects, *Bombyx mori* Linnaeus (Lepidoptera: Bombycidae), *Spodoptera exigua* (Hübner) (Lepidoptera: Noctuidae), and *Pieris rapae* (Linnaeus) (Lepidoptera: Pieridae). Also, in the ovarian calyx fluid of the endoparasitoid *Venturia canescens* (Gravenhorst) (Hymenoptera: Ichneumonidae), a serine protease was identified, showing the inhibition of melanization in the hemolymph of host *Ephestia kuehniella* (Zeller) (Lepidoptera: Pyralidae) [53].

The dehydrogenase class, whose role is also related to balancing and restoring physiological homeostasis, was also detected. This class of proteins also protects against environmental stresses, including oxidative ones, catalyzing the oxidation of toxic compounds [54,55]. Its presence among ovarian calyx fluid proteins could have a protective role for the initial necessary survival of the host, consequently guaranteeing the development of the parasitoid.

Among hydrolases, a trehalase-like protein was also found that generally has a role in carbohydrate metabolism in insects. This protein was also found to be highly expressed in the venom gland of several species of parasitoids, indicating a role in the complex host manipulations performed by parasitoid wasp venom [56]. In the host–parasitoid system *Spodoptera litura*–*Meterous pulchricornis*, it has been reported that the parasitoid induces the expression of two trehalases to ensure a suitable nutritional environment for parasitoid development [57].

Two interesting proteins are the enolase and the chitinase. Enolase, already found in *T. nigriceps* venom [23], is a multifunctional protein also released by teratocytes of *Aphidius ervi* (Haliday) (Hymenoptera: Braconidae). Enolase has been reported to be involved in the degradation of the extracellular matrix to provide food nutrition for the parasitoid larva [26,58]. Chitinase is an enzyme able to digest chitin, a component of the exoskeleton of arthropods [59]. It plays an important role in the physiology of different endoparasitoids, where teratocytes secrete chitinases to facilitate the emergence of larvae by degrading the host cuticle [60]. In the OPs, chitinase may carry out a similar action, assisting the parasitoid larva in hydrolyzing the host embryonic cuticle and reaching the host embryo. The chitinase from teratocytes of *T. nigriceps* was proposed for its potential to control *H. virescens*. Indeed, *T. nigriceps* chitinase was expressed into transgenic tobacco plants and the survival of larvae of *H. virescens* was severely affected when fed on leaves of transgenic tobacco expressing the recombinant chitinase [60,61].

Among kinases, phosphoglycerate kinase, involved in glycolysis and gluconeogenesis, and a hypothetical protein with a pyruvate kinase domain (all involved in energy pathways) were found. Pyruvate kinase is also an essential enzyme in the intracellular glucose metabolism of parasitic organisms [62]. Arginine kinase, an enzyme crucial for the energy metabolism of insects, was also detected, as previously found in the venom *Leptomastix dactylopii* (Howard) (Hymenoptera: Encyrtidae) [63].

Among OPs, ferritin was also identified: it is an intracellular protein that carries and stores iron. This protein was found in the venom gland of the endoparasitoid *Meteorus pulchricornis* (Wesmael) (Hymenoptera: Braconidae) [64] and of the ectoparasitoid *Torymus sinensis* (Kamijo) (Hymenoptera: Torymidae) [49], but its role is still unknown. Along with ferritin, transferrin, a multifunctional protein, could have several roles involving the subtraction of iron from the host; the protection of the parasitoid from oxidative stress induced by some host–immune responses [65]; and could support the success of oviposition by promoting the delivery of iron [66] and the development of oocytes [67].

Another protein that may have a double function, towards both host and parasitoid development, is the Krüppel homolog 1-like protein. Several studies reported the involvement of this protein in juvenile hormone signal transduction, to prevent larval–pupal metamorphosis in holometabolous species and nymphal–adult metamorphosis in hemimetabolous species [68,69]. The Krüppel homolog 1-like has been shown to be a transcriptional regulator factor that specifically binds to cis-regulatory elements in the promoters of genes encoding steroidogenic enzymes, inducing DNA methylation. This inhibits the transcription of steroidogenic enzymes and adversely affects the biosynthesis of ecdysone [70,71], essential for larval development [72,73]. Another function is related to vitellogenesis and egg development of the parasitoid; in many insect species, the blockage of egg development has been shown to occur through RNA interference [74,75,76,77,78,79].

A protein disulfide-isomerase, that catalyzes the oxidation of thiols and the isomerization of disulfides, was also identified. This protein was also found in the nematode *Heterodera schachtii* A.Schmidt (Rhabditida: Heteroderidae) as a putative effector that interferes with the host’s redox status. This protein could be involved in the observed oxidative stress in *T. nigriceps* hemocytes [35,36]. Also, glutaredoxins, similar to the glutathione transferase found in the active HPLC fraction #26, play a key role in maintaining the intracellular redox balance and protecting cells from oxidative damage that can be caused by host defense mechanisms [80,81].

Among the identified OPs, the presence of Alix, a protein involved in the apoptosis pathway, is very interesting. The *Tn*BV protein ANK1 has previously been shown to induce apoptosis in host hemocytes by interacting with Alix [32]. The presence of Alix among OPs is unexpected and shows how maternal factors could act in synergy to evade host immune defenses.

The HSP70 protein, HSP60 protein, and toll-like receptor 13 (TLR-13) were also found. The HSPs are a protein family of conserved ubiquitous proteins found in all living organisms. These proteins are chaperons involved in protein folding and help to protect cells from physiological stresses. In *G. mellonella*, the HSP70 was reported to be highly expressed after fungal infection [82], suggesting a cytoprotective role and an improvement of the resistance of the insect to infection.

The TLR-13 is an endosomal receptor that is not present in humans and it is activated by an unmethylated motif present in the large ribosomal subunit of bacterial RNA (23S rRNA). The presence of TRL-13 and HSP70 among the OPs confirms previous results [35]. The HSP70 and TRL-13 could be useful to prevent the attack of the host by other pathogens, because the parasitoid needs to inhibit the immune response of their host, but simultaneously needs to ensure the survival of the parasitoid larva until the end of development.

Finally, as previously said, components of the virus envelope, an egg yolk protein and a major royal jelly protein 5, were found, probably due to a contamination. Moreover, proteins involved in DNA replication (calcineurin and calmodulin), transcription (polymerase, replication factors), and protein synthesis (translation initiation factors, elongation factors) were identified. Other proteins identified were involved in lipid metabolism (such as mitochondrial lipoyl synthase and SEC14-like protein 2), together with other synthase molecules such as inositol-3-phosphate synthase, fundamental for the synthesis of many inositol-containing compounds, involved in the signaling transduction cascade of growth and hormonal regulation. Inositol-3-phosphate synthase could also play a role in the oxidative stress response, as reported for *Apis cerana cerana* (Fabricius) (Hymenoptera: Apidae) [83].

The proteasome non-ATPase regulatory subunit and proteasome subunits could help the parasitoid organism to clear the environment from host proteins, partially or completely degraded. Puromycin-sensitive aminopeptidase was also detected: this protein, strictly related to proteasome proteins, is the major peptidase responsible for digesting polyglutamine sequences released by proteasomes during protein degradation. Structural proteins such as actin, myophilin, or dymeclin, putatively involved in organizations in Golgi apparatus [84], and mitochondrial proteins, were also detected.

Our results allow a more comprehensive understanding of the components of ovarian calyx fluid proteins and also their possible role in synergy with other parasitic factors.

## 5. Conclusions

This study provides an overview of *T. nigriceps* OPs, identified by a combined transcriptomic and proteomic approach, and discusses their possible role in the parasitic syndrome. Although additional in vivo studies are needed, this is the first report on *T. nigriceps* ovarian calyx fluid protein composition and the results support the role of OPs in ensuring the success of parasitoidism, combined with other maternal factors.

Indeed, in the ovarian calyx fluid we have identified several potential candidates that could have possible roles in host immunosuppression and developmental regulation. In the following studies, their functional characterization will be necessary to confirm their possible role.

Moreover, our findings provide useful information for the study on the evolution of parasitoids. Indeed, some of the proposed functions discussed for *T. nigriceps* ovarian calyx proteins are similar to the function of gene products from many polydnaviruses, for example, the comparisons of ovarian calyx fluid protein sequences with the available sequences of polydnavirus proteins could be helpful for studies on their evolutionary history [85,86].

Finally, the study of host-parasitoid interactions by an omic approach, that allows the identification of host regulation factors of parasitic origin, provides useful information and molecules of natural origin that could be used in the biological control of insect pests. Although numerous other studies are necessary, several examples of factors of parasitic origin that have shown insecticidal activity are reported in the literature [47,61,87] and this could be true also for *T. nigriceps* OPs.

## Figures and Tables

**Figure 1 biomolecules-13-01547-f001:**
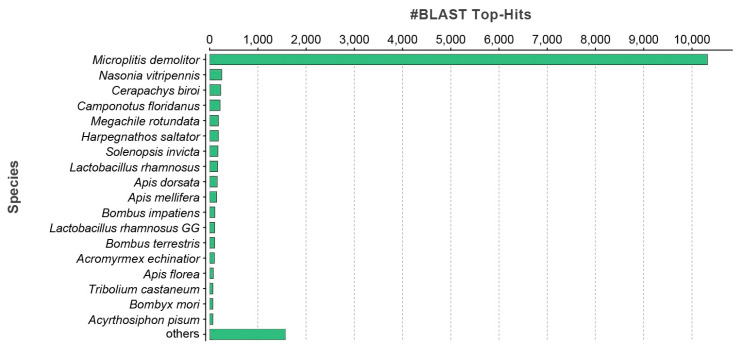
Top BLAST hit species for the transcriptome assembly of *T. nigriceps*. The x-axis shows the number of top BLAST hits for each species from the NCBI non-redundant protein database. The endoparasitoid wasp *Microplitis demolitor* obtained the highest number of matches.

**Figure 2 biomolecules-13-01547-f002:**
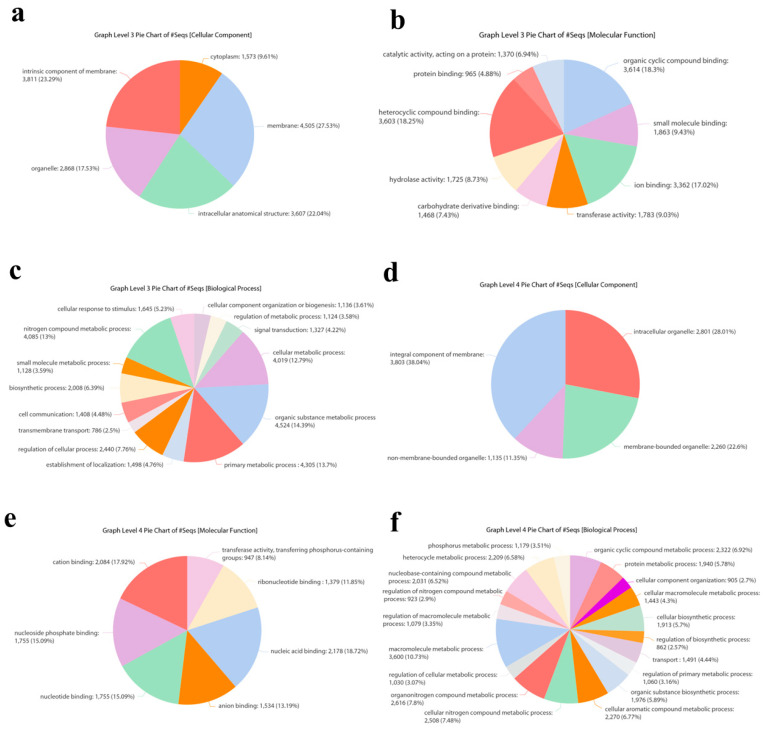
Gene ontology sequence annotation. Functional classification of the nr-matched transcripts of *T. nigriceps* ovarian calyx fluid proteins: (**a**,**d**), cellular components; (**b**,**e**), molecular functions; (**c**,**f**), biological processes. Data are presented as GO levels 3 and 4. Gene objects are shown as percentages of the total number of gene objects with GO assignments. Percentages below 2% were not reported.

**Figure 3 biomolecules-13-01547-f003:**
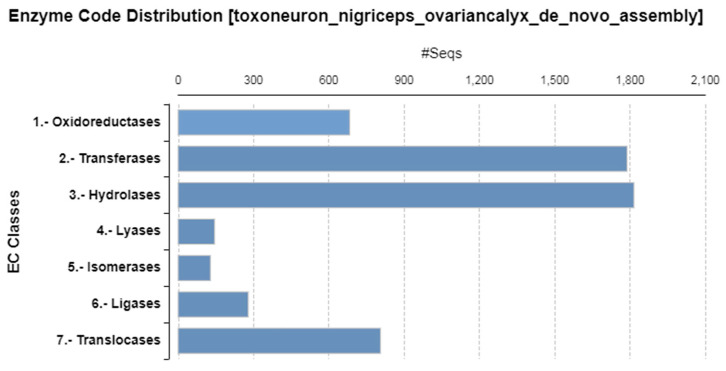
Distribution of enzyme code (EC) classes of contigs encoding enzymes in *T. nigriceps* ovarian calyx fluid proteins.

**Figure 4 biomolecules-13-01547-f004:**
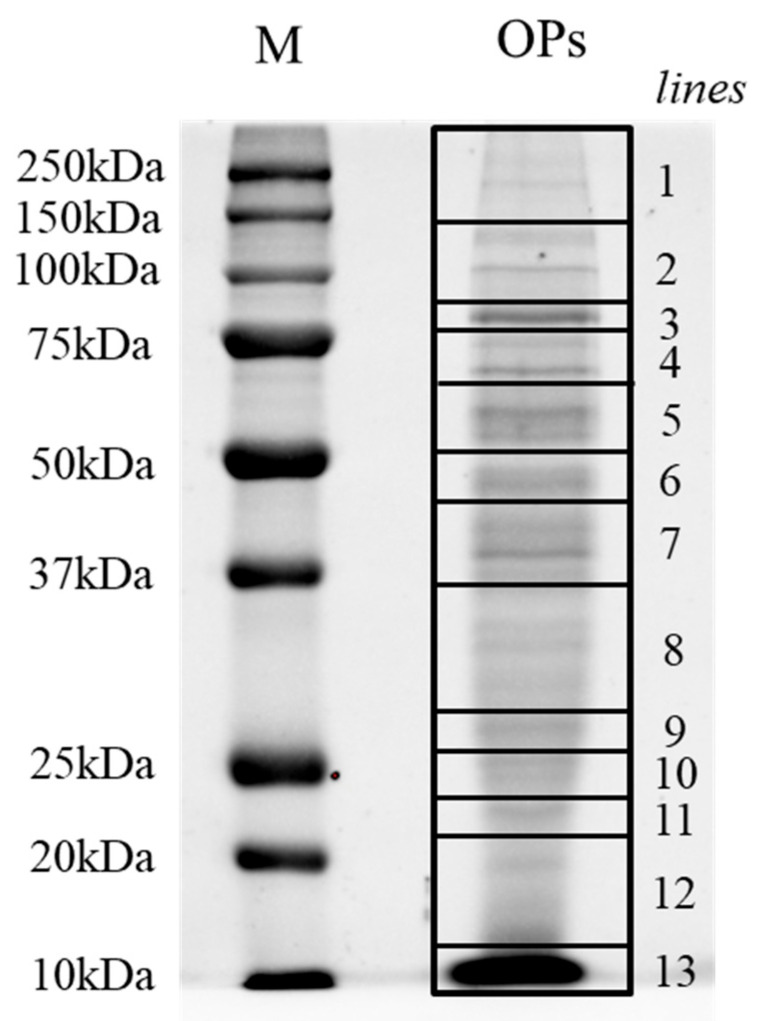
SDS-PAGE of ovarian calyx fluid proteins from *T. nigriceps* electrophoretic separation. The 13 lines indicated in the figure were excised and analyzed for protein identification. Lane M: molecular weight marker (expressed in kDa); lane OPs: ovarian calyx fluid proteins.

## Data Availability

The datasets used and/or analyzed during the current study are available from the corresponding author on reasonable request.

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
