# Peer review of "An Overview of Ovarian Calyx Fluid Proteins of Toxoneuron nigriceps (Viereck) (Hymenoptera: Braconidae): An Integrated Transcriptomic and Proteomic Approach"

_biomolecules, 2023, doi:10.3390/biom13101547_

Round 1
Reviewer 1 Report
Overall comment
The manuscript is fairly well written with an appropriate introduction informing the importance of this study. The methodology sounds and the results are believed to be very useful for future investigations and understanding of how parasitology is evolved.
However, I see that most of the results and discussion present in this manuscript are largely coming from the data generated by the proteome, while there is scarce information about how the transcriptomes generated in this study helps the reader to understand the molecular parasitoid function of Toxoneuron nigriceps. It is highly recommended that the authors could provide a deeper analysis on the transcriptomes to better link up the proteomic and transcriptomic data to provide a more integrated view, as mentioned in the title provided by the author. For details, please refer to the comments below.
Introduction
For line 93: “de novo” need to be italicsed.
Materials and Methods
1. Section 2.1: It would be nice if the authors could provide a brief description on the rearing method of Toxoneuron nigriceps, rather than directly referring that to another paper.
2. Section 2.2: Line 133: Please double check if the RNA was eluted in 20mL of storage solution. To me, the amount is a bit unreasonably large. Or is the unit a typo?
3. Section 2.2: Line 137: “de novo” need to be italicsed.
4. Section 2.3: Line 143: “appr.” is better to be written in a full word rather than an abbreviation.
5. Section 2.4: Line 182: Is the “ovarian calyx fluid proteins T. nigriceps database” the same as the one mentioned in Section 2.3 Line 160? I see both are databases for T. nigriceps but their names are different. If the authors are referring to the same database, please be more consistent in the naming and font format.
6. Section 2.4: How many biological and technical replicates were employed in the SDS-PAGE and mass spectrometry? Please specify.
Results
1. Line 193: Please correct the typing error statement “This The de novo transcriptome…”
2. Line 197: “BLASTx” is differently written as “blastx” in line 153 of materials and methods. Please be consistent with the naming of the algorithm.
3. Line 210-211: The letter C of “Cellular Component” should be written in small letters, which is more consistent to other categories.
4. Again, the database name and format of “Ovarian calyx fluid proteins T. nigriceps database” is not consistent throughout the whole manuscript.
Discussion
1. Line 292: “cleavage site” should be “cleavage sites”
2. Line 295: “whit” should be “with”
3. Line 295: Where were these homologues proteins found (from what specie) and how you determine their homology? Are there any phylogenetic trees supporting their phylogenetic gene homology?
4. Line 297: delete the redundant word “the”
5. Line 318: “2” should be written in “two”
6. Line 262: Please delete one redundant comma.
7. Line 373 & 376: “Krueppel homolog 1-like” should be written as “Krüppel homolog 1”
8. Since the authors have the transcriptome of the ovarian calyx, it would be more appropriate to compare the putative proteins identified in the transcriptomes, and the proteomes. A comparison should be made between the proteome and the transcriptomes to elucidate how many of the transcripts are actually translated into functional forms and how it inform us I the function of the organ and different components in the fluid.
Conclusion
1. I understand that the work here could potentially benefit the future invention of biopesticide. However, I don’t see any supporting evidence or discussion in which of the mentioned components in the ovarian calyx can be developed into a pesticide. In this context, I do not think it is appropriate to emphasise this aspect in the conclusion, or even in the introduction section. Unless the authors have any potential targets that they want to convey to the readers.
Figure
1. It would be more appropriate to italicise the species name in Figure 1.
2. T. nigriceps and Microplitis demolitor are needed to be italicised in the legend of all Figures.
3. The GO annotations for the pie charts in Figure 2 are way too small to read. Please enlarge the fonts to improve clarity and resolution.
4. It is highly encouraged to provide a table or chart to graphically display the components identified in the 13 bands of the PAGE gel. It would be more coherent to the discussion of different components in the discussion section.
Supplementary files
1. I see that supplementary file 3 includes the overall description of the other supplementary files. It is better to move this as the first file.
2. For supplementary file 2, please indicate and specify what are the red colours in the sequences.
The quality is fairly good to be understood from a reader's point of view.
Author Response
Comments and Suggestions for Authors
Overall comment
The manuscript is fairly well written with an appropriate introduction informing the importance of this study. The methodology sounds and the results are believed to be very useful for future investigations and understanding of how parasitology is evolved. However, I see that most of the results and discussion present in this manuscript are largely coming from the data generated by the proteome, while there is scarce information about how the transcriptomes generated in this study helps the reader to understand the molecular parasitoid function of Toxoneuron nigriceps. It is highly recommended that the authors could provide a deeper analysis on the transcriptomes to better link up the proteomic and transcriptomic data to provide a more integrated view, as mentioned in the title provided by the author. For details, please refer to the comments below.
We thank the reviewer for her/his comments and valuable suggestions that allow us to enrich our work by sharing a series of information present in the transcriptome and which we had underestimated in the first version of the work. Furthermore, his/her observation gives us the opportunity to better explain the approach used in our work in analogy to what was already done in the previous work done on another parasitic factor of T. nigirceps, the venom (Laurino et al. 2016). Also in that case, the identification of the venom proteins was carried out by a combined transcriptomic and proteomic approach since an annotated genome is not available for this insect and for the identification by proteomic analysis for our model system no sequences are available in database. Transcriptomic analysis of RNA obtained from ovarian calyx cells provides an overview of all transcripts of this tissue, which has a mechanical function of receiving the egg from the ovary and the function producing and releasing the proteic fluid named "ovarian calyx fluid", while proteomic analysis of ovarian calyx fluid, after separation from the Polydnavirus, gives direct information about the proteins actually expressed and secreted in the ovarian calyx lumen. Our work therefore focused on the unambiguous identification of the proteins of the calyx fluid, providing also an overall picture of the ovarian calyx transcriptome (see fig. 2 and 3) classifying the contigs into three Gene Ontology categories (cellular components, molecular functions and biological processes) and reporting the main enzyme families. However, following the reviewer's suggestion, to complete and provide a more integrated view of the transcriptome we added as supplementary material the file with the functional annotation of the 24,760 contigs of which the ovarian calyx is composed (supplementary material S2), in addition to the supplementary file S1 with the sequences in fasta format, which allows a more in-depth analysis of all the transcripts. The transcriptome has been further analyzed and contigs associated with the polydnavirus were also identified (table S3). In particular, we found 88 contigs, 16 belong to the wasp and annotated as viral capsid proteins but surprisingly we found also 72 sequences of viral genes normally expressed only into the host. However, we hypnotize that the presence of contigs related to the polydnavirus could due to residual polydnavirus DNA present in the RNA seq libraries, even if RNA was treated with Dnase, indeed the Rnaseq technique is so deep that can detect very tiny amounts of nucleic acids. We reported all this information in the revised manuscript.
Below the point-by-point responses to each of her/his corrections or suggestions are reported.
Introduction
For line 93: “de novo” need to be italicsed.
We thank the reviewer for pointing out the error, we corrected it.
Materials and Methods
- Section 2.1: It would be nice if the authors could provide a brief description on the rearing method of Toxoneuron nigriceps, rather than directly referring that to another paper.
We provided in the manuscript a brief description on the rearing method.
- Section 2.2: Line 133: Please double check if the RNA was eluted in 20mL of storage solution. To me, the amount is a bit unreasonably large. Or is the unit a typo?
We thank the reviewer for pointing out the typo, we corrected it.
- Section 2.2: Line 137: “de novo” need to be italicsed.
We thank the reviewer for pointing out the error, we corrected it.
- Section 2.3: Line 143: “appr.” is better to be written in a full word rather than an abbreviation.
We corrected it.
- Section 2.4: Line 182: Is the “ovarian calyx fluid proteins T. nigriceps database” the same as the one mentioned in Section 2.3 Line 160? I see both are databases for T. nigriceps but their names are different. If the authors are referring to the same database, please be more consistent in the naming and font format.
We thank the reviewer for his/her suggestion, we adjusted the database referring name indicating it as “ovarian calyx fluid proteins T. nigriceps database” in the main text.
- Section 2.4: How many biological and technical replicates were employed in the SDS-PAGE and mass spectrometry? Please specify.
For the SDS-PAGE we used a single sample from a pool of samples obtained from approximately 30 females. We provided this information in the materials and methods section of the revised manuscript.
Results
- Line 193: Please correct the typing error statement “This The de novo transcriptome…”
We thank the reviewer for pointing out the typo, we corrected it.
- Line 197: “BLASTx” is differently written as “blastx” in line 153 of materials and methods. Please be consistent with the naming of the algorithm.
We thank the reviewer for pointing out the error, we corrected it, in all the manuscript we wrote BLASTx.
- Line 210-211: The letter C of “Cellular Component” should be written in small letters, which is more consistent to other categories.
We thank the reviewer for her/his suggestion, we corrected it.
- Again, the database name and format of “Ovarian calyx fluid proteins T. nigriceps database” is not consistent throughout the whole manuscript.
According to the reviewer's suggestion, we modified the database name in all the text reporting “ovarian calyx fluid proteins T. nigriceps database”.
Discussion
- Line 292: “cleavage site” should be “cleavage sites”
We thank the reviewer for pointing out the error, we corrected it.
- Line 295: “whit” should be “with”
We thank the reviewer for pointing out the typo, we corrected it.
- Line 295: Where were these homologues proteins found (from what specie) and how you determine their homology? Are there any phylogenetic trees supporting their phylogenetic gene homology?
We thank the reviewer for this question. In the manuscript we are not implying orthology, but simply homology based on the best BLAST hits from the NCBI non-redundant (nr) database. The alignments to protein sequences from NCBI databases was mainly used to infer potential signal peptides for those transcripts that were 5´-end truncated. Homology as such does not require confirmation by gene trees (i.e. phylogenetic analyses), since this would only be required when implying orthology, for example when considering gene families.
- Line 297: delete the redundant word “the”
We thank the reviewer, we deleted it.
- Line 318: “2” should be written in “two”
We thank the reviewer, we corrected it.
- Line 262: Please delete one redundant comma.
We thank the reviewer, we deleted it.
- Line 373 & 376: “Krueppel homolog 1-like” should be written as “Krüppel homolog 1”
We thank the reviewer, we corrected it.
- Since the authors have the transcriptome of the ovarian calyx, it would be more appropriate to compare the putative proteins identified in the transcriptomes, and the proteomes. A comparison should be made between the proteome and the transcriptomes to elucidate how many of the transcripts are actually translated into functional forms and how it inform us I the function of the organ and different components in the fluid.
We thank the reviewer for her/his suggestion. The reviewer is right we have the entire transcriptome of the ovarian calyx tissue, that, as previously reported, has the mechanical function of receiving the egg from the overlying ovarioles and also has the role of producing and secreting the fluid of the ovarian calyx, that contains the Polydnavirus and a protein mixture. We thank the reviewer because her/his comment allows us to highlight from the transcriptome also genes of the polydnavirus, as we reported in the revised manuscript (supplementary material S3). Going back to what said above, regarding the transcriptomic analysis, we used the transcript sequences to generate a custom-made protein database. The six reading frames of the 24,760 contigs derived from the ovarian calyx transcriptome were translated in their respective putative amino acid sequences using SEQtools, obtaining 148,560 sequences (the “ovarian calyx fluid proteins T. nigriceps database”), the putative proteome. This database is very useful for protein identification after MS/MS analysis. Generally, the experimental results obtained from the mass spectrometry analysis (peak list) are compared with peak list available on online databases (freely available) using softwares such as Mascot (http:\\www.matrixscience.com). The peak list generated from our MS/MS results was uploaded in Mascot software, and we performed a research against the in silico peak list obtained from the custum - made database “ovarian calyx fluid proteins T. nigriceps database” that we generated since for our model system no sequences are available in database. The amino acid sequences identified by Mascot software allowed to uniquely identify the 105 ovarian calyx fluid proteins and their reading frames, and then the specific sequences were used for alignment with homologous proteins belonging to organisms phylogenetically close to the parasitoid, by using two different softwares, the BLASTx and the BLASTp to obtain the protein functional annotation.
Most of information were reported in results section but we provided some more information also in the discussion section in the revised manuscript.
Conclusion
- I understand that the work here could potentially benefit the future invention of biopesticide. However, I don’t see any supporting evidence or discussion in which of the mentioned components in the ovarian calyx can be developed into a pesticide. In this context, I do not think it is appropriate to emphasise this aspect in the conclusion, or even in the introduction section. Unless the authors have any potential targets that they want to convey to the readers.
The rewiever is absolutely right. With this work we added another piece to the puzzle of this complex system represented by the host-parasitoid interaction. This work represents the completion of two our previous papers in which the functional characterization of the entire ovarian calyx fluid secretion and of the main active fractions was performed (Salvia et al., 2021; Salvia et al., 2022). Indeed, in the first manuscript (Salvia et al., 2021) we have shown the macroscopic effects of Toxoneuron nigriceps ovarian calyx fluid proteins on host hemocytes. In particular, we observed that it induced several alterations on hemocytes, including cellular oxidative stress, modifications of actin cytoskeleton, vacuolization, and the inhibition of encapsulation capacity of hemocytes, thus inducing both a loss of hemocyte functionality and cell death. In the second paper on T. nigriceps ovarian calyx fluid proteins (Salvia et al., 2022), 2 HPLC fractions of ovarian calyx fluid proteins were found to be responsible for all the effects (oxidative stress, vacuolization, inhibition of encapsulation on haemocytes) observed in the previous work (Salvia et al., 2021) in which the entire ovarian calyx fluid was used. In addition, we identified the proteins of these two fractions by a combination of transcriptomic and proteomic approaches, resulting in the certain identification of eight proteins that could be involved in the observed alterations of the host hemocytes.
In the present work we have the certain identification of all ovarian calyx fluid proteins, thanks to mass spectrometry analysis, to provide a complete and overall picture of the components of this secretion which, similarly to other parasitic secretions, is characterized by a complex mixture of numerous proteins with different molecular weights and different concentrations, in which few proteins are often responsible for the most relevant alterations observed into the host. However, parasitic secretions are composed also by other proteins whose function requires more in-depth studies as they could be responsible for effects that are not immediately understandable and which often derive from synergistic actions with other parasitic factors.
However, having an overall picture of the components of the secretion characterized with functional annotation constitutes the basis for starting to evaluate the potential function in the parasitization, for which many studies are needed. The reviewer is absolutely right, indeed following her/his suggestion in the revised manuscript we removed this part and proceeded to broaden the discussion by emphasizing what for us are the most relevant results
Figure
- It would be more appropriate to italicise the species name in Figure 1.
We thank the reviewer for her/his suggestion, we modified the Figure 1.
- T. nigriceps and Microplitis demolitor are needed to be italicised in the legend of all Figures.
We thank the reviewer for her/his suggestion, we corrected it.
- The GO annotations for the pie charts in Figure 2 are way too small to read. Please enlarge the fonts to improve clarity and resolution.
The reviewer is absolutely right. We have increased the font size of Figure 2.
- It is highly encouraged to provide a table or chart to graphically display the components identified in the 13 bands of the PAGE gel. It would be more coherent to the discussion of different components in the discussion section.
We thank the reviewer for his/her suggestion, we added “Supplementary Table S4” with the 105 proteins identified in the 13 lanes of the proteomics analysis. The table reports only the contig code and the frame number with the relative number of identified peptides. We also reported the sequence coverage and the name of the identified protein. For all the other information relative to the peptide identification and the Blast results for each protein please refer to “Supplementary Table S5”.
Supplementary files
- I see that supplementary file 3 includes the overall description of the other supplementary files. It is better to move this as the first file.
The reviewer is right, we added a file word with all the supplementary material captions.
- For supplementary file 2, please indicate and specify what are the red colors in the sequences.
We better clarify the meaning of the red colors in the contig sequence by modifying the following sentence in the main text: “in red are reported the tryptic peptides experimentally observed identified by LC-MS/MS analysis, used for the sequence coverage calculation”. Also in the Table caption, in the revised manuscript now table S5, we specify the red colors meaning.
Reviewer 2 Report
Dear Authors and Editor,
The authors present a study on Toxoneuron nigriceps, an endoparasitoid insect targeting the tobacco budworm Heliothis virescens. The study investigates the venom and ovarian calyx fluid of T. nigriceps, particularly the ovarian calyx fluid proteins (OPs), which have received less attention than the venom components. The study's primary objective is to understand how these ovarian proteins affect the host's immune response and contribute to the parasitic process. The study aims to fill a gap in existing research by focusing on the less-studied ovarian calyx fluid proteins. This can lead to novel insights into the parasitic mechanisms of T. nigriceps and its interaction with its host.
Abstract - The abstract should explain the study's main findings and scientific implications. It should be rewritten entirely. The abstract clearly outlines the study's objective - to investigate the role of ovarian calyx fluid proteins in the parasitization process of T. nigriceps. It provides a concise context for the research. However, the abstract assumes a certain level of prior knowledge about Toxoneuron nigriceps, the tobacco budworm, and the parasitic relationship between the two species. For readers unfamiliar with these organisms, more context might be necessary to grasp the research's significance fully. Furthermore, the abstract mentions that the main components of ovarian proteins were identified, but it does not provide specific details about these components. This omission makes it difficult to gauge the extent of the findings.
Discussion: The beginning of the discussion describes elements of the methodology and results. While the authors mention that the study provides "useful information," they do not elaborate on the potential implications or applications of this information. What are the broader implications for parasitology or pest management? The authors do not emphasize any specific findings or outcomes of the study. Including even a brief discussion of the most significant results would provide a stronger sense of the study's contribution.
Minor concerns
L43 – Instead of victims, consider utilising prey or host;
L54-55 - Please avoid paragraphs consisting of a single sentence;
L59 - Please explain what kind of factors;
L285 – Plodydnavirus?
L289-291 - These sentences are combining methodology and results.
The text is easy to read and few English language errors were noticed.
Author Response
The authors present a study on Toxoneuron nigriceps, an endoparasitoid insect targeting the tobacco budworm Heliothis virescens. The study investigates the venom and ovarian calyx fluid of T. nigriceps, particularly the ovarian calyx fluid proteins (OPs), which have received less attention than the venom components. The study's primary objective is to understand how these ovarian proteins affect the host's immune response and contribute to the parasitic process. The study aims to fill a gap in existing research by focusing on the less-studied ovarian calyx fluid proteins. This can lead to novel insights into the parasitic mechanisms of T. nigriceps and its interaction with its host.
We thank the reviewer for her/his comments and valuable suggestions. Below the point-by-point answers to each of her/his corrections or suggestions are reported.
Abstract - The abstract should explain the study's main findings and scientific implications. It should be rewritten entirely. The abstract clearly outlines the study's objective - to investigate the role of ovarian calyx fluid proteins in the parasitization process of T. nigriceps. It provides a concise context for the research. However, the abstract assumes a certain level of prior knowledge about Toxoneuron nigriceps, the tobacco budworm, and the parasitic relationship between the two species. For readers unfamiliar with these organisms, more context might be necessary to grasp the research's significance fully. Furthermore, the abstract mentions that the main components of ovarian proteins were identified, but it does not provide specific details about these components. This omission makes it difficult to gauge the extent of the findings.
We really thank the reviewer for her/his suggestion. We have rewritten the abstract and we are confident that in this form it is clearer.
Discussion: The beginning of the discussion describes elements of the methodology and results. While the authors mention that the study provides "useful information," they do not elaborate on the potential implications or applications of this information. What are the broader implications for parasitology or pest management? The authors do not emphasize any specific findings or outcomes of the study. Including even a brief discussion of the most significant results would provide a stronger sense of the study's contribution.
We thank the reviewer for her/his suggestion, she/he is absolutely right. Following her/his suggestion we implemented the discussion by emphasizing the importance of our work and the most relevant results. We modified discussion and conclusion section emphasizing the importance of the identification of protein components of the fluid of the ovarian calyx of the parasitoid insect T. nigriceps. Indeed, to the best of our knowledge few studies of ovarian calyx fluid components of insect parasitoids are reported. Indeed although in the 2 previous works (Salvia et al. 2021 and 2022) we performed the functional characterization of the entire ovarian calyx fluid secretion (Salvia et al., 2021) and of the main active fractions (Salvia et al., 2022) showing the macroscopic effects of Toxoneuron nigriceps ovarian calyx fluid proteins on host hemocytes, with this work we added another piece to this complex puzzle of this complex system represented by the host-parasitoid interaction. The proteomic and transcriptomic analysis allowed us to identify all the proteins of the ovarian calyx fluid, including the less expressed ones. Our results provide useful information for subsequent studies on the mechanism of action of the identified proteins. Moreover, in the discussion and conclusion sections of the revised manuscript we reported examples of molecules effective in host regulation that have shown insecticidal activity on other insect pests
Minor concerns
L43 – Instead of victims, consider utilising prey or host;
We thank the reviewer for her/his suggestion, we corrected using host.
L54-55 - Please avoid paragraphs consisting of a single sentence;
We thank the reviewer for her/his suggestion, we corrected it.
L59 - Please explain what kind of factors;
We thank the reviewer for her/his suggestion, we provided information about the parasitic factors.
L285 – Plodydnavirus?
We thank the reviewer for pointing out the typo, we corrected.
L289-291 - These sentences are combining methodology and results.
We thank the reviewer for her/his suggestion, we modified the sentence.